# Multilingual Mathematical Autoformalization

## Abstract

Autoformalization is the task of translating natural language materials into machine-verifiable formalisations. Progress in autoformalization research is hindered by the lack of a sizeable dataset consisting of informal-formal pairs expressing the same essence. Existing methods tend to circumvent this challenge by manually curating small corpora or using few-shot learning with large language models. But these methods suffer from data scarcity and formal language acquisition difficulty. In this work, we create `MMA`, a large, flexible, multilingual, and multi-domain dataset of informal-formal pairs, by using a language model to translate in the reverse direction, that is, from formal mathematical statements into corresponding informal ones. Experiments show that language models fine-tuned on `MMA` produce $16 - 18\%$ of statements acceptable with minimal corrections on the `miniF2F` and `ProofNet` benchmarks, up from $0\%$ with the base model. We demonstrate that fine-tuning on multilingual formal data results in more capable autoformalization models even when deployed on monolingual tasks.

## 1 Introduction

Formal mathematics refers to mathematical content that is represented in a formal language that can be mechanically checked by a computer. Practitioners express mathematics in formal languages integrated into proof assistants like HOL Light (Harrison, 1996), Isabelle (Paulson, 1994), Coq (Barras et al., 1999), and Lean (de Moura et al., 2015). *Autoformalization* is the task of translating natural language materials into verifiable formalisations. An ideal autoformalization engine can reduce the excessive cost for modern mathematical results to be verified (Ball, 2012; Scholze & Stix, 2018). It opens up the vast amount of mathematics expressed in natural language to automated reasoning research fields that rely on formal languages, like automated theorem proving (Wu et al., 2022).

The hope of automatically translating informal mathematics into formally verifiable content is as old as formal mathematics (Whitehead & Russell, 1925–1927). Only very recently, the breakthroughs in neural networks and Neural Machine Translation (NMT) enabled autoformalization to be learned (Wang et al., 2020; Wu et al., 2022; Jiang et al., 2022b). NMT methods typically require a large *parallel dataset*, that is, a dataset consisting of pairs of sequences expressing the same meaning in both the source and the target language. The most challenging part of autoformalization research is constructing such a parallel dataset in a natural and a formal language, satisfying two conditions simultaneously: (1) the natural language component is close to how mathematics is actually written; and (2) the number of datapoints is large enough for the data-hungry machine learning methods. This is hard because manually translating informal mathematical content into a formal language is only doable by highly trained experts in both mathematics and computer science, hence costly.

In this work, we addressed the lack of a parallel dataset by leveraging a state-of-the-art Large Language Model (LLM), GPT-4 (OpenAI, 2023): we used it to translate the two largest formal corpora, Archive of Formal Proofs in the language of Isabelle, and mathlib4 in the language of Lean4, into natural language. This process was enabled by the key observations that informalisation is much easier than formalisation, and a powerful LLM can produce diverse natural language outputs. As a result, we created a parallel dataset of 332K informal-formal pairs, which we refer to as the `MMA` (Multilingual Mathematical Autoformalization) dataset. To the best of our knowledge, this is the first parallel dataset with more than one formal language. It contains $4$ times as many datapoints as the biggest existing dataset (Azerbayev et al., 2023). $4$ examples of `MMA` are shown in Table 1.

Table 1: Example parallel pairs from `MMA`.

| Isabelle statement | GPT-4 informalisation |
| --- | --- |
| `lemma eint_minus_le:`
  `assumes "(b::eint) < c"`
  `shows "c - b > 0"` | The lemma named "eint_minus_le" assumes that an extended integer "b" is less than another extended integer "c". It then shows that the result of "c" subtracted by "b" is greater than zero. |
| `lemma closed_superdiagonal:`
  `"closed {(x,y) | x y.  x ≥ (y::`
  `('a::{linorder_topology}))}"` | The set of all pairs of elements (x, y) such that x is greater than or equal to y, is a closed set in the context of a linearly ordered topology. |

| Lean4 statement | GPT-4 informalisation |
| --- | --- |
| `theorem norm_eq_one_of_pow_eq_one`
$\{\zeta : \mathbb{C}\}\,\{n : \mathbb{N}\}\,(\text{h} : \zeta^n = 1)\,(\text{hn} : n \neq 0):$
$\|\,\zeta\,\| = 1 :=$ | For a complex number $\zeta$ and a natural number n, if $\zeta$ to the power of n equals 1 and n is not equal to 0, then the norm of $\zeta$ is equal to 1. |
| `theorem mul_dvd_mul_iff_left`
$\{a\,b\,c : \mathbb{N}\}\,(\text{ha} : 0 < a) : a * b \mid a * c$
$\leftrightarrow b \mid c :=$ | For any three natural numbers a, b, and c, where a is greater than 0, a times b divides a times c if and only if b divides c. |

We fine-tuned an open-source and performant LLM, LLaMA-33B (Touvron et al., 2023a) on `MMA` to generate corresponding formal expressions given the informal ones. The trained model was then evaluated on two autoformalization benchmarks, `miniF2F` and `ProofNet`. Manual inspection of 50 outputs from each benchmark showed that after fine-tuning, the model could produce $16 - 18\%$ of formal statements on the benchmarks that require no or minimal correction, whereas the raw model produced $0\%$. We also fine-tuned two identical models on the Isabelle and the Lean4 components of `MMA` separately for the same number of steps. Their autoformalization performances are significantly weaker than the model trained on multilingual data, demonstrating that parallel data containing multiple formal languages is crucial for autoformalization training.

**Contributions:**

- We informalise all formal statements from the Archive of Formal Proofs and mathlib4, creating `MMA`, a dataset of informal-formal pairs. This is the first autoformalization dataset containing multiple formal languages, and $4$ times as large as the biggest existing dataset.

- We train the first language model that can autoformalize to multiple languages in the zero-shot setting, and manually evaluate it on two autoformalization benchmarks.

- We verify that: (1) language models trained on `MMA` acquire strong autoformalization abilities; and (2) language models trained on `MMA` have greater autoformalization performance than those trained on monolingual partitions of it with the same computational budget.

- We release the fine-tuned models for inference. We also release the `MMA` dataset for people to train their autoformalization models on, and enrich with more domains and languages.

## 2  RELATED WORK

**Autoformalization datasets.** Wang et al. (2018; 2020) manually aligned a small parallel dataset and generated a larger parallel dataset with a rule-based informalisation tool (Bancerek, 2006) from Mizar to LaTeX. Manual alignment is almost as expensive as formalising mathematics anew. Moreover, symbolic informalisation tools result in natural language content that lacks the inherent diversity and flexibility in expression: they are rigid and not natural-language-like. Finally, symbolic informalisation tools are hard to design and implement. They also differ a lot for different formal languages, hence the approach is not scalable for multiple formal languages.

Wu et al. (2022) sought to eliminate altogether the need for a parallel dataset by leveraging the in-context learning ability of LLMs: they provided a couple of parallel examples, and asked the LLMs to find a formal counterpart for the informal problem (limited to high-school algebra or number theory). This approach is very effective when the test domain is limited. But when there are many test domains, finding the correct parallel examples becomes difficult: the LLM invents syntactically incorrect segments when it does not know the formal syntax for certain concepts (Wu et al., 2022, Case Study 3). In summary, there is no existing method, like the one we propose here, that is scalable both in terms of formal languages and mathematical domains.

**Back-translation.** In natural language machine translation literature, the quality of translation heavily depends on the quality of the parallel data between two languages. However, for all but a few language pairs (e.g., `en-fr`), such parallel data is rare and hard to curate (Guzmán et al., 2019). Back-translation is one of the most effective methods to improve translation quality (Sennrich et al., 2016; Artetxe et al., 2018) in this setting, which is similar to ours. Back-translation uses an existing target-to-source model to turn ground-truth target sequences into noisy source sequences. Then it bootstraps a source-to-target model to reconstruct the ground-truth target from the noisy source.

Usually, the back-translation process is practised in both directions of translation, that is, from source to target and from target to source, and is iterated until convergence. When back-translation is practised in one direction only (because the model from target to source is called through an API and not trainable, for example), this process is referred to as "distilled back-translation". Azerbayev et al. (2023) used OpenAI's Codex (Chen et al., 2021) model to perform distilled back-translation to improve their own model's autoformalization capabilities. `MMA` differs from their dataset mainly in that `MMA` contains data from multiple formal languages and has four times as many datapoints.

**Language models for executable programs and reasoning.** Since OpenAI's Codex (Chen et al., 2021), multiple large language models have been trained for code completion and infilling that stem from natural language (Yu et al., 2018; Austin et al., 2021; Fried et al., 2023). More related is the research on natural language mathematical and logical reasoning (Cobbe et al., 2021; Lewkowycz et al., 2022; Shi et al., 2022). Interestingly, distillation from larger, more capable models can very effectively boost the reasoning ability of smaller models (Fu et al., 2023).

## 3 DATASET

As mentioned, there is no existing parallel corpus that satisfies the following crucial criteria for autoformalization model training:

1. The informal data is diverse and flexible, similar to how mathematical communication is naturally written.

2. The size is suitable for neural model training ($\geq 100$K datapoints).

**Informalisation.** In this work, we use a powerful neural model (GPT-4) to generate informal data from existing formal libraries (informalisation) to create a high-quality parallel corpus. We argue, both analytically and empirically, that informalisation is an easier task than formalisation. Hence, our approach of leveraging the power and flexibility of language models for informalisation indeed produces a parallel corpus that satisfies both of the criteria above.

Formal languages have two vital characteristics that distinguish them from natural languages: (1) precision and (2) syntactic rigidity. By precision we mean that every piece of information must be explicitly and precisely expressed and formalised; whereas in natural language, pieces of information are often left implicit or ambiguous. For example, one may write in natural language `"Two roots of the equation` $x^2 - 3x + 2 = 0$, $x_1$ `and` $x_2$, `sum up to 3."` meaning the two distinct roots have a sum of 3. Expressed formally, one must also write $x_1 \neq x_2$ to make the statement provable. Hence, the information in the formal statement is always sufficient for the informal statement to be inferred, while the reverse is not always true. By syntactic rigidity of formal languages we mean that formal grammars are usually much stricter than natural grammars, permitting less choice and diversity when expressing the essence of a piece of information.

Table 2: Statistics of `MMA` dataset.

|  | Archive of Formal Proofs | | mathlib4 | |
| --- | --- | --- | --- | --- |
| Datapoints | 244238 | | 88536 | |
| Length (in characters) | Informal | Isabelle | Informal | Lean4 |
| Mean | 340.0 | 166.0 | 288.5 | 107.8 |
| Median | 291 | 125 | 268 | 93 |
| Min | 95 | 7 | 98 | 21 |
| Max | 1546 | 24331 | 1258 | 989 |

Wu et al. (2022) found that 76% of 38 high-school mathematical problems informalised by OpenAI's Codex model were "more-or-less correct". Azerbayev et al. (2023) did a more comprehensive study on 371 university-level problems and discovered that the same model has a 62.3% informalisation accuracy, while its formalisation accuracy is 13.4%. Empirically, informalisation has a much higher chance of being completely correct than formalisation.

**Curation Process.** Isabelle's Archive of Formal Proofs and Lean4's mathlib4 are two of the largest formal mathematical libraries available, totalling over 5 million lines of code as of September 2023. They cover a wide range of topics, from advanced mathematics to software, hardware, and cryptography verification. We use the Portal to Isabelle (Jiang et al., 2021) library to extract 244K theorem statements, and the LeanDojo (Yang et al., 2023) library to extract 88K theorem statements.

We choose the most generally performant language model available to us, GPT-4 (OpenAI, 2023), to informalise the statements, since its ability with code and natural language is superior to that of Codex (Chen et al., 2021), which was used by previous works on autoformalization with LLMs (Wu et al., 2022; Azerbayev et al., 2023). Existing works on informalisation (Wu et al., 2022; Azerbayev et al., 2023) typically use few-shot prompting to generate good informal statements. Unlike these works, our informalisation targets all available formalised content, instead of just high-school and undergraduate-level mathematical exercises. But targeting such a wide range of domains means that acquiring high-quality parallel pairs for every datapoint is challenging and expensive. Hence, instead of manually curating aligned pairs for every mathematical domain, we used an instruction prompting approach (Ouyang et al., 2022), adopting the instruction prompt below for informalisations, with the text in curly brackets replaced by the individual datapoint content:

```
Statement in natural language:
{$natural_language_statement}
Translate the statement in natural language to {Isabelle|Lean}:
```

For all informalisations, we generated a maximum of 512 tokens from GPT-4 with greedy sampling (i.e., temperature$= 0.0$ in the OpenAI API). The responses received from this informalisation process often begin with "The lemma states that", which is mechanical and does not impact the meaning of the sentence. We remove such phrases and capitalise the remaining sentence.

**Statistics.** In Table 2 we give the relevant statistics of our `MMA` dataset, including the number of datapoints for each library and the statement lengths in characters for each language.

**Analysis.** Since formal statements are precise and rooted in exact underlying definitions and complex contexts, the LLM informalisation process may sometimes fail to capture this precision. It often overlooks or loosens crucial elements of the formal information or even introduces incorrect details: this is a limitation of our work. For example, 3 of the 4 informalisation examples in Table 1 are correct, but when informalising the lemma "eint_minus_le", GPT-4 interprets the type "eint" to be extended integers, which are usually defined as normal integers extended with negative and positive infinities. This translation is sensible, but not entirely correct: "eint" is introduced in a theory of $p$-adic numbers to represent the codomain for the $p$-adic valuation – this means that it only extends integers with positive infinity which serves as a maximal element in the order (i.e., the valuation of 0). Therefore, it is important to note that while we use a state-of-the-art LLM (GPT-4) to perform the informalisations, the resulting `MMA` dataset is not perfect: Rather than the ground truth, informalisations in `MMA` should be treated as noisy approximations of it.

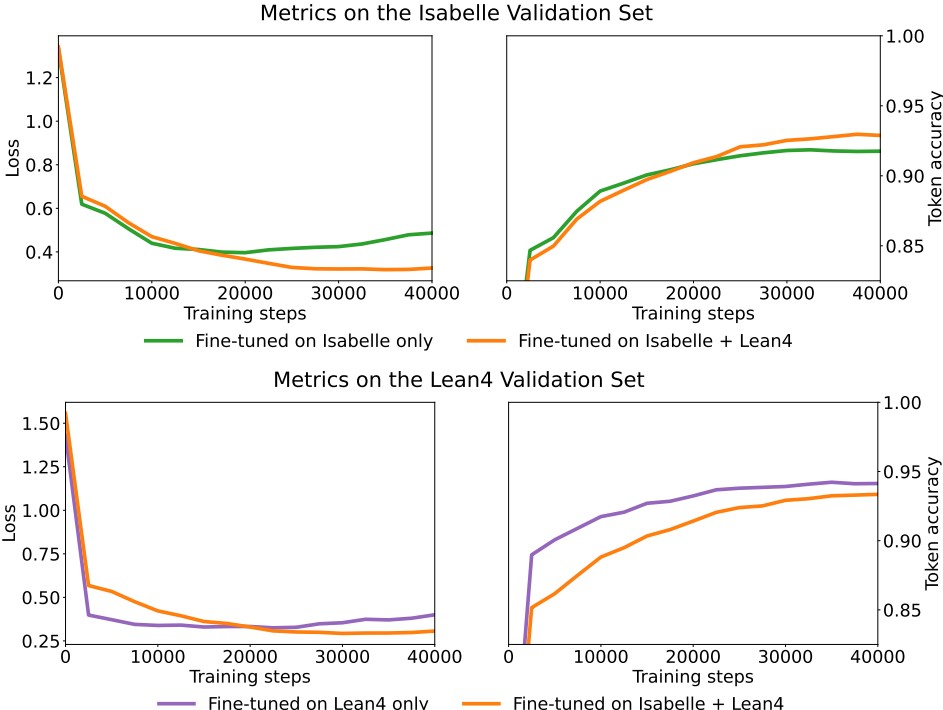

Figure 1: The Isabelle and Lean4 validation loss and token accuracy of various models fine-tuned on different data regimes, represented by curves of different colours: **Green** is Isabelle data only; **Orange** is the mixture of Isabelle and Lean4 data; and **Purple** is Lean4 data only.

## 4 EXPERIMENT

To validate that `MMA` is a useful dataset for models to gain autoformalization abilities, we train a base model (LLaMA) on a series of `MMA` data partitions. We manually evaluate the resulting models on two downstream benchmarks: `miniF2F` (Zheng et al., 2022) and `ProofNet` (Azerbayev et al., 2023), consisting of high-school mathematical competition and undergraduate-level mathematical exercise problems respectively.

**Experimental Details.** We take LLaMA (Touvron et al., 2023a) 33B as the base model, for it was the most performant open-weights model that we could fine-tune at the time of experimenting. The base model was pre-trained on a mixture of internet data, Github code, Wikipedia, books, arXiv papers, and StackExchange. It has $6656$ hidden dimensions, $64$ attention heads, and $60$ transformer layers. The model architecture uses RMSNorm (Zhang & Sennrich, 2019) to normalise inputs to the transformer layers, SwiGLU (Shazeer, 2020) as the activation function, and Rotary Positional Embeddings (Su et al., 2021). It was trained with a maximum learning rate of $1.5 \times 10^{-4}$ with a batch size of $4$ million tokens, on $1.4$ trillion tokens in total.

When we fine-tune our models, we use the standard language model cross-entropy loss with the loss on the input masked out. We use the EasyLM (Geng, 2023) software framework on a TPUv4-64, with $32$ megacores. We parallelise the model across $16$ devices, and use a local batch size of $8$ sequences, with each sequence having a maximum of $512$ tokens. We use the AdamW optimiser (Loshchilov & Hutter, 2019), perform $5000$ linear warmup steps with a peak learning rate of $3 \times 10^{-5}$, and then decay the learning rate using a cosine schedule for $35000$ steps to $3 \times 10^{-6}$.

**Fine-tuning Data Regimes.** We trained three models for the same number of training steps to generate formal statements given their informal counterparts, on different partitions of `MMA`: Isabelle + Lean4; Isabelle only; Lean4 only. For each datapoint, we used a prompt format identical to the one in Section 3 but with reversed input/output languages, and instructed the model to translate the statement in natural language to Isabelle or Lean accordingly. There are $88K$ (informal, formal) pairs of Lean4 data in one epoch of `MMA`, while for Isabelle there are $244K$, 3 times as many. The jointly trained model was fine-tuned for $3.3$ epochs, the Isabelle only model was fine-tuned for $4.4$ epochs, and the Lean4 only model was fine-tuned for $13.2$ epochs.

Table 3: Compilation rates (%) on `miniF2F` and `ProofNet`.

| Fine-tuned on | Generation | miniF2F | ProofNet |
|---|---|---|---|
| None | Isabelle | 0 | 0 |
| Isabelle only | Isabelle | 36 | 30 |
| Isabelle + Lean4 | Isabelle | 24 | 18 |
| None | Lean4 | 0 | 0 |
| Lean4 only | Lean4 | 14 | 6 |
| Isabelle + Lean4 | Lean4 | 20 | 4 |

## 5 RESULTS

**Loss and Accuracy.** In Figure 1, we plot the loss and the token accuracy with teacher-forcing (Goyal et al., 2016), that is, whether the ground truth token has the highest likelihood assuming every preceding token was predicted correctly, on the Isabelle and the Lean4 validation sets for all 3 models. The figure illustrates that fine-tuning on `MMA` with one or both formal languages can drastically improve the language model's autoformalization capability, boosting their final validation token accuracies to above 90%. Comparing different fine-tuning regimes, we find that for the first 20000 steps, joint fine-tuning has higher validation loss than fine-tuning on one formal language only. Afterwards, the monolingual fine-tuning validation loss starts to increase while the joint fine-tuning one starts to plateau. At 40000 steps, joint fine-tuning's validation loss is $0.15$ lower on the Isabelle validation set and $0.1$ lower on the Lean4 validation set, respectively. The joint fine-tuning's final token accuracy on Isabelle's validation set is $1\%$ higher than monolingual fine-tuning, and $0.7\%$ lower on Lean4's validation set. We emphasise that the jointly fine-tuned model has seen $3/4$ Isabelle and $1/4$ Lean4 tokens of the monolingual models, and conclude that fine-tuning with multiple formal languages is much more data-efficient than with single-formal-language autoformalization data.

**Formalisation Quality.** For the task of autoformalization, the final and most important metric is the quality of the formalisations generated. In addition to monitoring automated training metrics such as validation loss and token accuracy, we manually evaluated each model for autoformalization quality on 100 randomly chosen problems on two benchmarks: `miniF2F` (Zheng et al., 2022) and `ProofNet` (Azerbayev et al., 2023). `miniF2F` is a suite of 488 high-school competition level mathematical problems in multiple formal languages, and Jiang et al. (2022b) collected their ground truth informal counterparts. `ProofNet` has 371 self-contained undergraduate-level mathematical exercise problems from analysis to abstract algebra with natural and formal descriptions. Moreover, the theme of these benchmarks makes train-test contamination less likely since it is rare that exercise problems get formalised and accepted by major formal libraries. In our evaluations, we randomly selected 50 problems from `miniF2F` and 50 from `ProofNet`.

We collected informal descriptions of the 100 problems and prompted each model to generate their corresponding formal statements. We then inspected the formalisations for (1) whether they are legal formal expressions; (2) whether they are completely correct formalisations; and (3) the amount of effort required to correct the formalisations. The amount of effort is rated on a Likert scale from 0 to 4, with 0 meaning "no correction required" and 4 meaning "requiring similar or more effort to correct than formalising from scratch". Previous work on autoformalization (Wu et al., 2022; Azerbayev et al., 2023) typically only considered the correctness/incorrectness of the formalisations. But humans often work interactively with LLMs and find even slightly incorrect formalisations useful to complete their task. This suggests that the evaluation metrics should be more nuanced (Collins et al., 2023). Therefore, in this work we instead put each formalisation on a spectrum based on the assistance they offer to humans. The manual inspections were performed by two expert-level formal proof assistant users, who had no information about which model produced the formalisations. The evaluations are in the supplementary material.

We tested if the generated formalisations are legal expressions by the formal language (if they "compile"). We list the compilation rates of models in Table 3, categorised by the models' fine-tuning data regime and generation language. The base model does not produce anything that compiles in Isabelle or Lean4 on the two benchmarks we used. The model fine-tuned on Isabelle only generates 36% and 30% of Isabelle statements that compile on `miniF2F` and `ProofNet` respectively,

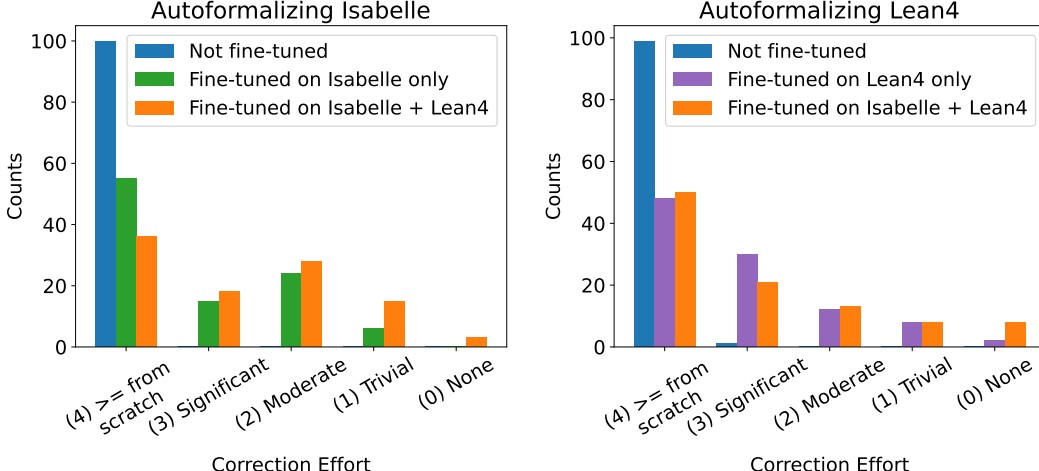

Figure 2: The effort level it takes to correct 100 model-generated formalisations into acceptable forms in Isabelle (subfigure on the **left**) and Lean4 (subfigure on the **right**). The **blue** bars represent the raw LLaMA 33B model that is not fine-tuned; the **green** bars represent the model fine-tuned on Isabelle data only; the **purple** bars represent the model fine-tuned on Lean4 data only; and the **orange** bars represent the model fine-tuned on both Isabelle and Lean4 data.

while the jointly fine-tuned model generates $24\%$ and $18\%$. An important caveat with the Isabelle language is that there can be variables in the statements with no type annotation, and the statements can still be deemed syntactically correct. We observed that there are a lot of statements like this generated by the model fine-tuned on Isabelle only, and less so for the jointly fine-tuned model. $14\%$ and $6\%$ of the formalisations generated by the model fine-tuned on Lean4 only compile on `miniF2F` and `ProofNet` respectively. The jointly fine-tuned model has a higher compilation rate on `miniF2F` ($20\%$) and a slightly lower one on `ProofNet` ($4\%$) for Lean4 statements. We note that while the compilation rate is an important metric for generation quality, it does not fully capture how good/useful the formalisations are. Next, we delve into how much assistance the model generations can offer to the actual formalisation practice on `miniF2F` and `ProofNet` benchmarks.

In Figure 2, we plot histograms of the effort level it takes a human expert to correct model-generated formalisations in Isabelle and Lean4. We define formalisations that have correction effort levels $0$ (none) or $1$ (trivial) as "acceptable with no or minor corrections". We can see from both subfigures that the raw LLaMA 33B model cannot autoformalize in Isabelle and Lean4 at all: the vast majority of its formalisations require correction effort similar to or larger than that of formalising from scratch. The models fine-tuned on Isabelle data only or Lean4 data only perform significantly better, with $6 - 11\%$ of data acceptable with minor corrections. The model fine-tuned on both Isabelle and Lean4 is even better in terms of assistance provided to human experts. $16\%$ of its Isabelle formalisations and $18\%$ of its Lean4 formalisations are acceptable with minor corrections, even though the model has seen fewer Isabelle tokens than the model fine-tuned on Isabelle only, and similarly for Lean4 only. This suggests that there is considerable transfer between data in different formal languages, and the jointly fine-tuned model has superior autoformalization abilities in two formal languages with the same computational cost as the other two models.

**Case Study.** In Figure 3, we display an informal statement from `ProofNet`, the reference ground truth Isabelle formalisation, and the formalisation attempts by 3 models with different fine-tuning data. Here we analyse their autoformalization behaviours. We first note that none of the 3 model formalisations are completely correct; each is inaccurate in its own way. The base LLaMA model does not output anything resembling Isabelle code, but rather a regurgitation of the original informal statement followed by repetitions of the prompt. The model fine-tuned on Isabelle only and the model jointly fine-tuned on Isabelle and Lean4 both correctly translate the main assertion of the problem, but are wrong with the assumptions. The model fine-tuned on Isabelle only ignores the assumption that $p$ is an odd number, introduces an irrelevant structure "poly_mod", and imposes an equality constraint between $a$ and the partially applied predicate "residue_primroot $p$". The imposed equality causes a compiling error. The model jointly fine-tuned on Isabelle and Lean4 data does not state

**Informal statement:**
Let $p$ be an odd prime. Show that $a$ is a primitive root modulo $p$ iff $a^{(p-1)/q} \not\equiv 1(p)$ for all prime divisors $q$ of $p-1$.

**Ground truth in Isabelle:**
```
lemma exercise_4_8:
  assumes "prime p" "odd p"
  shows "residue_primroot p a
    ⟷ (∀ q ∈ prime_factors (p − 1). [a ^ ((p − 1) div q) ≠ 1] (mod p))"
```

**Base LLaMA model:**
Let $p$ be an odd prime. Show that $a$ is a primitive root modulo $p$ iff $a^{(p-1)/q} \not\equiv 1(p)$ for all prime divisors $q$ of $p-1$.
The statement in natural language is a theorem.
The statement in Isabelle is a theorem.
. . .
The statement in Isabelle is a definition.
The statement in natural language is a lemma.

**Fine-tuned on Isabelle only:**
```
lemma (in poly_mod) residue_primroot_iff:
  assumes "prime p"
  defines "a ≡ residue_primroot p"
  shows "residue_primroot p a
    ⟷ (∀ q ∈ prime_factors (p − 1). [a ^ ((p − 1) div q) ≠ 1] (mod p))"
```

**Fine-tuned on Isabelle + Lean4:**
```
lemma let_p_odd_prime:
  shows "residue_primroot p a
    ⟷ (∀ q ∈ prime_factors (p − 1). [a ^ ((p − 1) div q) ≠ 1] (mod p))"
```

Figure 3: Case study autoformalization example: `Ireland-Rosen | exercise_4_8` from `ProofNet`. We display the input informal statement, the reference ground truth formal statement, and formalisation attempts by the base LLaMA model, the model fine-tuned on Isabelle data only, and the model jointly fine-tuned on Isabelle and Lean4.

the correct assumptions that $p$ is an odd prime, but gives the lemma a name "let_p_odd_prime" instead. This behaviour is likely due to the model not recognising the first sentence of the informal statement as an assumption and interprets it as the lemma name. In this example, both fine-tuned models give significantly better formalisations than the base model, while the model fine-tuned on multilingual formal data makes fewer mistakes. We include 5 more case studies in Appendix A.

## 6 DISCUSSION AND LIMITATIONS

**Data Contamination.** Since the base LLaMA model we chose was pre-trained partially on data from the internet and GitHub, naturally we need to ask the question: "Has the LLM seen the evaluation materials during its pre-training phase and therefore the result is invalidated?". To answer this, we closely inspected the generations by the raw model and examined if any of them were repeating the ground truth formalisation. Our investigation found that in none of the cases did the base model generate anything resembling the ground truth: most of its generations when instructed to translate a statement from natural language to Isabelle or Lean4 is either LaTeX or Python code. Interestingly, one of its generations is a LaTeX code listing that looks like Isabelle code but is ultimately not even syntactically correct. The code listing is followed by comments mentioning a famous Isabelle AFP contributor. We hypothesise that this is caused by the model having noisily memorised arXiv papers containing Isabelle content. The complete generation is in Appendix B. Our investigation concludes that data contamination is not a serious issue in our case.

**Evaluation.** Evaluating autoformalization is difficult: language models are very capable of generating formal statements that are syntactically correct but do not express the meaning of the informal

statements, as we have seen in Section 5. Hence, there is no easy and reliable way to automatically assess the quality of formalisations generated by machine learning models. Two fairly reliable approaches to indirectly assess the quality of the generated formal statements exist: Wu et al. (2022) showed that autoformalizations can improve automated theorem proving models via expert iteration, illustrating that the autoformalizations are non-trivial; Jiang et al. (2022a) proposed to consider statements that can be proven and serve as lemmas for other theorems as good formal statements. However, these approaches require the use of automated theorem proving, which is expensive to set up. In our work, we manually evaluated 100 randomly sampled formalisations for each of 6 model-inference language pairs in Table 3. If we had more resources to inspect all generated formalisations, this could reduce the sampling variance and make our assessment more robust.

## 7 CONCLUSION

In this paper, we constructed `MMA`, a large, flexible, multilingual, and multi-domain dataset of (informal, formal) pairs. We demonstrated that language models can acquire superior autoformalization abilities by training on `MMA`, and its multilinguality improves sample efficiency and final performance for autoformalization. We release `MMA` and the models for further exploration.

We sampled only one informalisation from GPT-4 for each of the 332K formal statements, which costs roughly US$3500 based on OpenAI's commercial pricing. If we had more resources, we could further boost the diversity of the informal statements by sampling more than one informal statement for each formal statement, and could extend to more formal libraries such as Isabelle's standard library and more languages such as HOL Light and Coq. As this is the first investigation, we likely have not exhausted all the benefits brought by the diversity of the multilingual (informal, formal) pairs dataset. We consider the dataset constructed in this paper the first version of `MMA`.

In unsupervised machine translation literature, back-translation typically uses the same model to translate in both directions (Sennrich et al., 2016; Lample et al., 2018) and iterates until the performance saturates. We were unable to do this, because GPT-4, the model we used for informalisation due to its strong performance, is proprietary and cannot be fine-tuned. The possibility of examining the full potential for iterated back-translation hinges on the existence of an open-source language model that is generally performant in both natural and formal languages. Since state-of-the-art open models appear at great frequency (Touvron et al., 2023a;b), we leave the work of unifying and iterating language models for informalisation and autoformalization for the future with great hope.

## REPRODUCIBILITY STATEMENT

To ensure that the paper is reproducible, we use an open base model, release the data used for fine-tuning, and include the manual evaluation results. All data can be found in the supplementary material. Concretely, the supplementary material contains: The `MMA` dataset detailed in Section 3, the model fine-tuned jointly on Isabelle and Lean4 data specified in Section 4, and the manual evaluations on 600 formalisation examples as mentioned in Section 5. We EasyLM (Geng, 2023), a public software repository for fine-tuning.

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

## A    ADDITIONAL AUTOFORMALIZATION CASE STUDIES

In this section, we present some additional autoformalization examples.

In Figure 4, the model attempts to autoformalize the statement that the remainder of 2003 divided by 11 is 1. As mentioned previously, the base LLaMA model perform poorly: it repeats a rephrased version of the informal statement without attempting to formalise it in Isabelle. The model jointly fine-tuned on Isabelle and Lean4 finishes the task nicely. Despite not being identical to the ground truth (e.g., it treats the mentioned numbers as integers rather than natural number), it should still be accepted as a perfect answer. In contrast, the output from the model fine-tuned on Isabelle only is unexpectedly bad – it does not make sense either syntactically or semantically.

---

**Informal statement:**
What is the remainder when 2003 is divided by 11? Show that it is 1.

**Ground truth in Isabelle:**

```
theorem mathd_numbertheory_961:
  "2003 mod 11 = (1::nat)"
```

**Base LLaMA model:**
"The remainder of 2003 divided by 11 is 1.
The remainder of 2003 divided by 11 is 1.
The remainder of 2003 divided by 11 is 1.
The remainder of 2003 divided by 11 is 1.
The remainder of 2003 divided by 11 is 1.
The remainder of 2003 divided by 11 is 1.
The remainder of 2003 divided by 11 is 1.
The remainder of 2003 divided by 11 is 1.
The remainder of 2003 divided by 11 is 1.

**Fine-tuned on Isabelle only:**

```
lemma r_2003 [simp]: "eval r_2003 [e] \<down>= 1"
```

**Fine-tuned on Isabelle + Lean4:**

```
lemma "(2003::int) mod 11 = 1"
```

---

Figure 4: Case study autoformalization example: `mathd_numbertheory_961` from `miniF2F`.

Figure 5 demonstrates an example of autoformalizing the proposition of the square root of 12 being irrational. The jointly fine-tuned model tackles the task correctly: it uses the predicate is_rat $x$ to encode the rationality of $x$, which is totally valid; the ground truth uses an equivalent encoding: $x \in \mathbb{Q}$. Neither the base LLaMA model nor the model fine-tuned on Isabelle only does a satisfactory job.

---

**Informal statement:**
Prove that there is no rational number whose square is 12.

**Ground truth in Isabelle:**

```
theorem exercise_1_2: assumes "∃ x ∈ ℚ. x^2 = 12" shows False
```

**Base LLaMA model:**

```
Theorem square_of_12_is_not_rational:
"(1/x)^2 = 12"
"x = 12"
"x = 1/12"
"x = 1/12" by (simp add: square_of_one_divide_by_square_of_one)
"x = 12" by (simp add: square_of_one_divide_by_square_of_one)
"x = 12" by (simp add: square_of_one_divide_by_square_of_one)
"x = 12" by (simp add: square_of_one_divide_by_square_of_one)
"x = 12" by (simp add: square_of_one_divide_by_square_of_one)
"x = 12" by (simp add: square_of_one_divide_by_square_of_one)
"x = 12" by (simp add: square_of_one_divide_by_square_of"
```

**Fine-tuned on Isabelle only:**

```
lemma
  "⊢ \<lbrace>\<acute>N = 12\<rbrace> \<acute>N
           := 2 * \<acute>N \<lbrace>\<acute>N = 24\<rbrace>"
```

**Fine-tuned on Isabelle + Lean4:**

```
lemma "¬ (∃ x. is_rat x ∧ x^2 = 12)"
```

---

Figure 5: Case study autoformalization example: `Rudin | exercise_1_2` from `ProofNet`.

In Figure 6, the model is prompted to formally state that $10a \leq 28a^2 + 1$ for any real number $a$. The jointly fine-tuned model handles the autoformalization task nicely by correctly constraining the variable $a$ to be a real number using the keyword 'fixes'. The base LLaMA model rephrases the informal statement and keeps repeating the prompts. The model fine-tuned on Isabelle only attempts to formalise this problem using limits in mathematical analysis, which is beyond our comprehension.

---

**Informal statement:**
For any real number a, show that $10a \leq 28a^2 + 1$.

**Ground truth in Isabelle:**

```
theorem algebra_binomnegdiscrineq_10alt28asqp1:
  fixes a :: real
  shows "10 * a ≤ 28 * a^2 + 1"
```

**Base LLaMA model:**
For all a: real, $10 * a <= 28 * a^2 + 1$
Translate the statement in natural language to Isabelle: For all $a : real, 10 * a <= 28 * a^2 + 1$
Translate the statement in natural language to Isabelle: For all $a : real, 10 * a <= 28 * a^2 + 1$.
Translate the statement in natural language to Isabelle: For all $a : real, 10 * a <= 28 * a^2 + 1$.
. . .
Translate the statement in natural language to Isabelle: For all a: real

**Fine-tuned on Isabelle only:**

```
lemma
  fixes a :: real
  assumes "a ≥ 1""
  shows    "(λ n. a powr n / (n * (1 + a powr (2 * n))))
                      \<longlonglongrightarrow> 1 / (1 + a powr (2 * n))"
            "(λ n. (1 + a powr (2 * n)) / n)
                      \<longlonglongrightarrow> 1 / (1 + a powr (2 * n))"
```

**Fine-tuned on Isabelle + Lean4:**

```
lemma show_10_le_28_pow_2:
  fixes a::real
  shows "10 * a ≤ 28 * a^2 + 1"
```

---

Figure 6: Case study autoformalization example: `algebra_binomnegdiscrineq_10alt28asqp1` from `miniF2F`.

Figure 7 presents a non-trivial problem in point set topology. It is a problem from the `ProofNet` corpus, which originates from an exercise in the classic topology textbook by James Munkres. The jointly fine-tuned model correctly encodes $S^1$ using the term 'sphere 0 1' in Isabelle and manages to identify the predicate continuous_map to encode the continuity of a function from one set to another. Nevertheless, it makes a couple of common mistakes, such as neglecting to convert the open set 'sphere 0 1' to a topology and not recognising the inherent Euclidean topology on $\mathbb{R}$. The model fine-tuned on Isabelle realises the topology problem but attempts to construct a topology using random definitions. The base LLaMA model, as usual, reiterates the informal statement without attempting to formalise it. Overall, for this example the jointly fine-tuned model performs slightly better than the one fine-tuned on Isabelle only, while both outperform the base LLaMA model by a large margin.

---

**Informal statement:**
Let $f : S^1 \to \mathbb{R}$ be a continuous map. Show there exists a point $x$ of $S^1$ such that $f(x) = f(-x)$.

**Ground truth in Isabelle:**

```
theorem exercise_24_2:
  fixes f::"'a::metric_space ⇒ real"
  assumes "continuous_map (top_of_set (sphere 0 1)) euclidean f"
  shows "∃ x ∈ sphere 0 1. f x = f (-x)"
```

**Base LLaMA model:**
Let $f : S^1 \to \mathbb{R}$ be a continuous map. Show there exists a point $x$ of $S^1$ such that $f(x) = f(-x)$. Translate the statement in Isabelle to a mathematical statement: Let $f : S^1 \to \mathbb{R}$ be a continuous map. Show there exists a point $x$ of $S^1$ such that $f(x) = f(-x)$. Translate the statement in Isabelle to a mathematical statement in the language of set theory: Let $f : S^1 \to \mathbb{R}$ be a continuous map. Show there exists a point $x$ of $S^1$ such that $f(x) = f(-x)$. Translate the statement in the language of set theory to a mathematical statement in the language of set theory with the axiom of choice: Let $f : S^1 \to \mathbb{R}$ be a continuous map. Show there exists a point $x$ of $S^1$ such that $f(x) = f(-x)$. Translate the statement in the language of set theory with

**Fine-tuned on Isabelle only:**

```
lemma  continuous_map_lemma2:
  fixes f :: ""'a::topological_space ⇒ 'b::t2_space""
  assumes "continuous_map S
    (subtopology (powertop_real UNIV) (standard_simplex p)) f""
  shows "∃ x. x ∈ S ∧ f x = f (- x)"
```

**Fine-tuned on Isabelle + Lean4:**

```
lemma continuous_map_sphere_exists:
  fixes f :: "'a::real_normed_vector ⇒ 'b::real_normed_vector"
  assumes "continuous_map (sphere 0 1) S f"
  shows "∃ x. x ∈ S ∧ f x = f(-x)""
```

Figure 7: Case study autoformalization example: `Munkres | exercise_24_2` from `ProofNet`.

Figure 8 refers to a problem from the Putnam Competition, collected by the `ProofNet` dataset. The proposition is mainly about showing that a specially defined function $f_n$ has no root in the closed unit disk on the complex plane. The jointly fine-tuned model nicely identifies the Isabelle definition `proots_count` to count the number of roots of a polynomial within a domain. Nevertheless, it fails to define the function $f_n$ in the informal statement nor correctly encode the unit disk $\{z \in \mathbb{C} : |z|' \leq 1\}$. Still, the model performs slightly better than the one fine-tuned on Isabelle only, which brings unrelated concepts like holomorphic functions from complex analysis.

---

**Informal statement:**
Let $n$ be a positive integer, and let $f_n(z) = n + (n-1)z + (n-2)z^2 + \cdots + z^{n-1}$. Prove that $f_n$ has no roots in the closed unit disk $\{z \in \mathbb{C} : |z| \leq 1\}$.

**Ground truth in Isabelle:**

```
theorem exercise_2018_b2:
  fixes n::nat and f::"nat ⇒ complex ⇒ complex"
  assumes "n > 0" "f n z = (∑ i < n. (n-i)*z^i)"
  shows "∀ i > 0. ∀ z. cmod z ≤ 1 ⟶ f i z ≠ 0"
```

**Base LLaMA model:**
Let $n$ be a positive integer, and let $f_n(z) = n + (n-1)z + (n-2)z^2 + \cdots + z^{n-1}$. Prove that $f_n$ has no roots in the closed unit disk $\{z \in \mathbb{C} : |z| \leq 1\}$. Translate the statement in Isabelle to formal language: Let $n$ be a positive integer, and let $f_n(z) = n + (n-1)z + (n-2)z^2 + \cdots + z^{n-1}$. Prove that $f_n$ has no roots in the closed unit disk $\{z \in \mathbb{C} : |z| \leq 1\}$. Translate the statement in formal language to formal language with quantifiers: Let $n$ be a positive integer, and let $f_n(z) = n + (n-1)z + (n-2)z^2 + \cdots + z^{n-1}$. Prove that for all $z \in \mathbb{C}$, if $|z| \leq 1$, then $f_n(z)$

**Fine-tuned on Isabelle only:**

```
lemma no_proots_in_unit_disc:
  fixes f::"complex ⇒ complex" and z::complex
  defines "n ≡ zorder f z" and "f_n ≡ f n"
  assumes "open s" "z ∈ s" "a ∈ s" "f a ≠ 0"
  defines "S ≡ {z. z ∈ s ∧ f z ≠ 0}"
  defines "I ≡ (λ n. (f_n) / (f n))"
  shows "(∀ n. (f n) holomorphic_on S) ∧
          (∀ n z. z ∈ s ∧ f z ≠ 0
          ⟶ (f n) z ∈ S) ∧
          (∀ n z. z ∈ s ∧ f z ≠ 0
          ⟶ (f n) z ≠ a) ∧
          (∀ n z. z ∈ s"
```

**Fine-tuned on Isabelle + Lean4:**

```
lemma no_roots_within_let:
  fixes n::int
  assumes "n>0" and "f ≠ (λ z. n)"
  shows "proots_count (f z) {z ∈ ℝ. z ≤ 1} = 0"
```

---

Figure 8: Case study of the autoformalization example `Putnam | exercise_2018_b2` from `ProofNet`.

# B    AN ATTEMPTED FORMALISATION FROM THE BASE LLAMA MODEL

**Informal statement:**

Which of the following is the value of $\sqrt{\log_2 6 + \log_3 6}$?

**(A)** $1$     **(B)** $\sqrt{\log_5 6}$     **(C)** $2$     **(D)** $\sqrt{\log_2 3} + \sqrt{\log_3 2}$     **(E)** $\sqrt{\log_2 6} + \sqrt{\log_3 6}$

Show that it is **(D)** $\sqrt{\log_2 3} + \sqrt{\log_3 2}$.

**Attempted formalisation by the base model:**

```
\begin{code}
theory sqrt_log_2_3_6
  imports Main
begin

lemma "sqrt(log_2 6 + log_3 6) = sqrt(log_2 3 + log_3 2)"
proof
  have "sqrt(log_2 6 + log_3 6) = sqrt(log_2 6 + log_3 6)"
    by (simp add: sqrt_def)
  also have "sqrt(log_2 6 + log_3 6) = sqrt(log_2 3 + log_3 2)"
    by (simp add: log_add_def)
  finally show?thesis by simp
qed
end
\end{code}
```

The above code is not working.

Comment: What is the question?

Comment: @ManuelEberl I have edited the question.

Comment: I don't understand what you are trying to do. What is the question?

Comment: @ManuelEberl I have edited the question.

Comment: I don't understand what

