# OpenReview forum: "Multilingual Mathematical Autoformalization"
_ICLR.cc/2024/Conference — Submitted to ICLR 2024_

### Official Review · Reviewer_rYoT · 2023-10-30

**Soundness:** 3 good
**Presentation:** 3 good
**Contribution:** 2 fair
**Rating:** 3
**Confidence:** 4

**Summary:**

This work translates formal problem descriptions into informal descriptions in natural language with GPT-4, and demonstrates on two standard benchmarks that the collected data is helpful to improve autoformalization.

**Strengths:**

Significant improvement of the performance on autoformalization.

**Weaknesses:**

The scope and novelty is limited.
This paper can be viewed as a back-translation version of Fu et al. (ICML 2023; see below for reference details).
The proposed method is generic, but only evaluated on the narrow domain of autoformalization.

### Missing References
Three lines of related work are almost completely missing.
Representative work in each line is listed below.
Please conduct a literature search using the papers below as starting points and cite relevant papers in their references.
1. **Model distillation** \
    Fu et al. Specializing Smaller Language Models towards Multi-Step Reasoning. ICML 2023

2. **Mathematical and logical reasoning with LLMs, and related work that involves multiple (natural) languages** \
    Cobbe et al. Training verifiers to solve math word problems. 2021 arXiv preprint: 2110.14168 \
    Lewkowycz et al. Solving Quantitative Reasoning Problems with Language Models. NeurIPS 2022 \
    Shi et al. Language Models Are Multilingual Chain-of-Thought Reasoners. ICLR 2023

3. **Executable program as formalisms of natural language** \
    In addition to OpenAI's Codex, the following work is also worth checking:\
    Yu et al. Spider: A large-scale human-labeled dataset for complex and cross-domain semantic parsing and text-to-SQL task. EMNLP 2018 \
    Austin et al. Program synthesis with large language models. 2021 arXiv preprint: 2108.07732 \
    Fried et al. InCoder: A Generative Model for Code Infilling and Synthesis. ICLR 2023

### Minor Comments and Typos
Term consistency needs double check: for example, both *formalisation* and *formalization* appear in the paper.

**Questions:**

- Have you also checked potential data contamination issues? For example, would it be possible that GPT-4 has seen the test data in the evaluation benchmarks?

---

> ### Author Response · Authors · 2023-11-17
> **Official rebuttal (part 1)**
>
> We thank the reviewer for helpful suggestions and references. Below we address the specific points raised by the reviewer.
>
> - Novelty:
>   - As we specify in the Contributions subsection, the main contributions of this paper includes:
>     1. Creating a large dataset
>     2. Training multiple language models on it with different settings
>     3. Evaluating these language models on held-out benchmarks manually to demonstrate that the dataset helps autoformalization
>     4. Conducting ablation experiments to show the multilinguality in formal languages benefits autoformalization.
>
>   - We want to emphasise that this is a dataset paper, not a method one. The paper by Fu et. al proposes “model specialization”, which distils from a large model to a small model. Ours is about utilising the power of model distillation and back-translation to construct the most useful dataset for autoformalization, and validating various scientific points with it, as mentioned in the contributions.
>
>   - > The proposed method is generic, but only evaluated on the narrow domain of autoformalization.
>
>      Autoformalization is a new domain that poses unique challenges like the lack of a sizeable dataset, the integration of neural and symbolic methods, and the difficulty of model evaluation. We think the domain is important enough to warrant its own treatment: Numerous papers solely on the specialised domain of autoformalization have been published, with lots in top conferences within both the machine learning and the theorem proving communities [1-8].
>
>   - Back-translation and sequence distillation are well-known techniques in the field of NLP and we do not make claims about inventing them. This is a dataset paper and we utilise these two techniques to create the dataset. The advantage of using back-translation for autoformalization is that informalising is much easier than formalising and hence we can get a high-quality dataset with it. Only the very first part of the process and experiments described in our paper is similar to Fu et. al and our novelty is elsewhere.
>
> - References:
>   - We note that autoformalization is quite different from both program synthesis and informal reasoning, with the aim of translating from a natural language to a formal one instead of creating formal programs from scratch. These papers are related to the method of creating the dataset, rather than the conclusions/contributions (which are about the dataset and the multilinguality) of the paper. We thank the reviewer again for the references. We have updated the paper to include these references and made a comparison of our paper to them.
>
> - Data contamination:
>   - This is a valid concern and we appreciate it. Three important pieces of information indicate that there is unlikely any data contamination with our paper:
>     - We did not evaluate GPT-4. We only evaluated llama models.
>     - The pre-trained model is not contaminated: We dedicate a subsection to the issue of data contamination with the pre-trained llama model in Section 6. Since it cannot even be instructed to write Isabelle or Lean, we think it unlikely that the pre-trained model is contaminated.
>     - The fine-tuning datasets are not contaminated: The fine-tuning datasets come from Isabelle AFP + Lean Mathlib4. The test benchmarks come from miniF2F + ProofNet. These datasets are strictly disjoint. We use GPT-4 to create English descriptions of AFP and Mathlib4, which are extremely unlikely to include anything from the formal test benchmarks.
> Since we have an unlikely contaminated pre-trained model, and an unlikely contaminated fine-tuning dataset, we conclude that the test evaluations are not affected by data contamination, and our conclusions stand.
>
> - Spelling:
>   - We use British English spelling throughout, except for the word “autoformalization”, which has become a term.

---

> > ### Author Response · Authors · 2023-11-17
> > **Official rebuttal (part 2)**
> >
> > [1] Wang, Qingxiang, et al. "Exploration of neural machine translation in autoformalization of mathematics in Mizar." Proceedings of the 9th ACM SIGPLAN International Conference on Certified Programs and Proofs. 2020.
> >
> > [2] Wu, Y., Jiang, A.Q., Li, W., Rabe, M., Staats, C., Jamnik, M. and Szegedy, C., 2022. Autoformalization with large language models. Advances in Neural Information Processing Systems, 35, pp.32353-32368.
> >
> > [3] Hahn, Christopher, et al. "Formal specifications from natural language." arXiv preprint arXiv:2206.01962 (2022).
> >
> > [4] Reichel, Tom, et al. "Proof Repair Infrastructure for Supervised Models: Building a Large Proof Repair Dataset." 14th International Conference on Interactive Theorem Proving (ITP 2023). Schloss-Dagstuhl-Leibniz Zentrum für Informatik, 2023.
> >
> > [5] Jiang, A.Q., Welleck, S., Zhou, J.P., Lacroix, T., Liu, J., Li, W., Jamnik, M., Lample, G. and Wu, Y., 2022, September. Draft, Sketch, and Prove: Guiding Formal Theorem Provers with Informal Proofs. In The Eleventh International Conference on Learning Representations.
> >
> > [6] Cosler, Matthias, et al. "nl2spec: Interactively Translating Unstructured Natural Language to Temporal Logics with Large Language Models." arXiv preprint arXiv:2303.04864 (2023).
> >
> > [7] Xin, Huajian, et al. "LEGO-Prover: Neural Theorem Proving with Growing Libraries." arXiv preprint arXiv:2310.00656 (2023).
> >
> > [8] First, Emily, et al. "Baldur: whole-proof generation and repair with large language models." arXiv preprint arXiv:2303.04910 (2023).

---

> > > ### Author Response · Authors · 2023-11-20
> > >
> > > Dear Reviewer rYoT,
> > >
> > > Did our rebuttal sufficiently address your concerns? Is there anything we can present that will convince you to increase your rating? We look forward to hearing from you.
> > >
> > > Many thanks,
> > > Authors

---

### Official Review · Reviewer_1cNi · 2023-11-01

**Soundness:** 4 excellent
**Presentation:** 3 good
**Contribution:** 3 good
**Rating:** 8
**Confidence:** 4

**Summary:**

This paper describes the creation of a parallel dataset for informal natural-language mathematical statements and their formal counterparts in two proof-assistant languages. The dataset, the largest of its kind, is creating using the technique of back-translation. The paper then uses this dataset to fine-tune a LLM on the task of formalizing natural-language statements, which improves its performance a lot (compared to 0%).

**Strengths:**

This is an important problem, and the dataset will be helpful for many others working in this area. The dataset is the largest of its kind by far.

In the introduction, the fact that the dataset covers multiple formal languages does not necessarily sound like a selling point to me, until you point out that training a model on multiple formal languages helps. I think this is a strong point that could be emphasized more (if I understand it correctly, see below under Questions).

**Weaknesses:**

I feel that the word "Multilingual" is confusing because it sounds like it works on multiple natural languages. However, I admit that "multi-formal-language" is awkward.

The data is automatically generated by back-translation ("informalization"). This is good because informalization appears to be an easier task than formalization and because it's more important for the target (formal language) side of the data to be high quality. However, it also does mean that the source (natural language) side of the data might not be correct or might not be as realistic as it could be. As the authors note, "the resulting MMA dataset is not perfect: Rather than the ground truth, informalisations in MMA should be treated as noisy approximations of it."

**Questions:**

It's possible that I misunderstood the claim that training on mixed-language data helps. I did not understand the sentence "We emphasise that the jointly fine-tuned model has seen 3/4 Isabelle and 1/4 Lean4 tokens of the monolingual models, and conclude that fine-tuning with multiple formal languages is much more data-efficient than with single-formal-language autoformalization data" (p.6). The jointly fine-tuned model has a greater total amount of data, right? How many steps are in an epoch? By 40000 steps, has the jointly fine-tuned model seen more examples than the Lean4-only and Isabelle-only models?

---

> ### Author Response · Authors · 2023-11-17
> **Official rebuttal**
>
> We thank the reviewer for their insightful feedback. We are glad that the reviewer recognises the contribution of the paper. Below, we respond to specific questions:
>
> - The use of the word multilingual:
>   - We think it’s an appropriate word, since there are multiple formal languages. However we do admit that there can be confusion in this word. We will make the best attempt to contextualise it by making clear that the languages refer to formal languages whenever we use the word “multilingual”
>
> - Dataset quality
>   - We thank the reviewer for this comment. Indeed the dataset is noisy since it was generated by GPT-4. Hence it warrants an in-depth analysis, which we perform in the overall comment: https://openreview.net/forum?id=QqdloE1QH2&noteId=Z8PeCPad8b
>   - To summarise the gist of the analysis:
>     - Around 75% of the examined 200 problems were informalised completely correctly by GPT-4.
>     - The failure cases are mostly due to ambiguity from the lack of appropriate context. And to ameliorate this issue is an exciting avenue for future research, possibly leveraging retrieval-augmented generation.
>   - Moreover, in the paper, we demonstrate with comprehensive experiments that the dataset can be used in practice to boost the autoformalization ability of language models; and the ablation experiments highlight the importance of the multilinguality of formal languages.
>
> - Clarification of the mix-data training
>   - For all fine-tuned language models, we perform 40000 steps of training. Each model was trained on 40000 * 8 (local batch size) * 2 (world size) * 512 (sequence length) ~= 328 million tokens (containing pad tokens). Each model has seen an equal number of tokens.
>     - For the model trained on the mixture of two formal languages, one epoch is 332K sequences. It was trained for 3.3 epochs (~246 million Isabelle tokens, 82 million Lean4 tokens).
>     - For the model trained on Isabelle only, one epoch is 244K sequences. It was trained for 4.4 epochs (~328 million Isabelle tokens).
>     - For the model trained on Lean4 only, one epoch is 88K sequences. It was trained for 13.2 epochs (~328 million Lean4 tokens).

---

> > ### Comment · Reviewer_1cNi · 2023-11-17
> > **"Multilingual"**
> >
> > It's not terribly scientific, but it appears that in the world of compilers, "multi-language compiler" is far more common than "multilingual compiler" as measured by Google hits (172M vs 4M).

---

> > > ### Author Response · Authors · 2023-11-17
> > >
> > > That's a very good point! We shall rename it to multi-language when the word "multilingual" can cause confusion. Thank you very much for the comment!

---

### Official Review · Reviewer_Gtv6 · 2023-11-01

**Soundness:** 2 fair
**Presentation:** 3 good
**Contribution:** 4 excellent
**Rating:** 6
**Confidence:** 3

**Summary:**

The work introduces an automated dataset for mathematical formalization, composed of informal-formal pairings. This dataset was generated using a reverse translation approach on two formal corpora, utilizing the large language model (LLM). Experimental results from two benchmark tests (100 examples) supports the enhancements achieved through fine-tuning on this synthesized dataset. A notable discovery from the study is that fine-tuning on a multilingual (formal language) dataset can also yield benefits for a monolingual (formal language) benchmark.

**Strengths:**

* This work is the first effort in distilling mathematical formalization from the LLM, a contribution given the notable scarcity of parallel datasets of mathematical formalization.

* Both the generated dataset and the fine-tuned model will be made publicly available for further research and development.

**Weaknesses:**

* The evaluation could benefit from further enhancement. Currently, a sample size of only 50 examples from the benchmark is used, which may not provide sufficiently convincing results due to potential statistical limitations.

* As the author also mentioned, the synthesized datasets could be noisy. It would be beneficial to also include GPT4 in your comparative analysis for formalization quality. This would offer deeper insights into the noise levels within the generated datasets, i.e., how noisy it could be.

* The term ‘language’ or ‘lingual’ as used throughout the paper, especially in the section of introduction, could potentially lead to confusion, although it is understandable with careful reading.

**Questions:**

* Considering that the total number of examples required for complete testing is approximately **eight times** the number of the sampled 50 examples used in the evaluation, I would appreciate more information about the overall effort required to execute the complete testing process.

---

> ### Author Response · Authors · 2023-11-17
> **Official rebuttal**
>
> We thank the reviewer for their insightful feedback. We are glad that the reviewer recognises the contribution of the dataset given the scarcity of mathematical autoformalization data. Below we respond to specific points raised by the reviewer:
>
> - Evaluation:
>   - The main conclusions can be supported by 100 datapoints per model/inference configuration
>     - We sample evaluation datapoints randomly (not cherry-picked)
>     - With this much data, we already demonstrate very strong differences between the models to support our conclusions (16-18% vs 6-11% vs 0% of trivial corrections).
>   - It’s incredibly expensive to annotate the entire benchmarks in such detail
>     - We would like to point out that for this paper, we manually examine 50 examples from each benchmark x 2 benchmarks x 6 different model/language inference settings = 600 individual examinations. Evaluating each example requires 2-10 minutes for expert-level formal mathematicians, and in practice it took ~30 expert hours to finish. This is quite expensive. To execute the complete testing process, it requires 8-12x more effort, hence roughly 240-360 expert hours. This is USD 12K - 18K at a rate of $50/hr for formal mathematics experts (>= PhD level).
>   - However, we do understand that a sample size of only 50 examples per benchmark (10% of the entire dataset) is smaller than traditionally done. Therefore to address the issue of low sample size, since we release the evaluation datapoints, we will open the evaluation up for the Isabelle and Lean communities. We have already identified Isabelle and Lean experts who are willing to contribute to this annotation endeavour. Leveraging the power of open-source communities is the best way to scale the evaluations while maintaining the quality of them.
>
> - MMA dataset quality
>   - Please see the overall comment for an in-depth analysis of the quality of the MMA dataset: https://openreview.net/forum?id=QqdloE1QH2&noteId=Z8PeCPad8b
>   - To summarise the findings of the analysis:
>     - Around 75% of the examined 200 problems were informalised completely correctly by GPT-4.
>     - The failure cases are mostly due to ambiguity from the lack of appropriate context. And to ameliorate this issue is an exciting avenue for future research, possibly leveraging retrieval-augmented generation.
>
> - Confusion about the use of “language”
>   - When we use the word “language”, we try to make sure that it has relevant context around it to distinguish between a natural or a formal language.

---

> > ### Comment · Reviewer_Gtv6 · 2023-11-17
> >
> > Thanks for the explanation. It makes sense to me that the evaluation is very expensive at this moment and it would be better to ask help from the community.
> >
> > I’m glad that you took the suggestion from Reviewer 1cNi to use multi-language instead of multilingual. That would definitely improve the readability of the paper

---

> > > ### Author Response · Authors · 2023-11-17
> > >
> > > Thank you for your reply!
> > >
> > > Have our rebuttal and additional analysis adequately addressed your concerns? What do you think would it take at this discussion stage to further improve the quality of the paper?

---

> > > > ### Comment · Reviewer_Gtv6 · 2023-11-21
> > > >
> > > > Although I accept it would be hard for you to finish a complete evaluation by the end of the discussion period, I still have concern about the evaluation in the current version. Clarification itself is not enough for me to change the estimation. I'd like to keep my score at the moment.

---

### Official Review · Reviewer_d1G8 · 2023-11-02

**Soundness:** 3 good
**Presentation:** 3 good
**Contribution:** 2 fair
**Rating:** 5
**Confidence:** 4

**Summary:**

Summary
This study centers on the autoformalization of natural language into machine-verifiable formalizations, employing a back-translation approach with GPT-4 to convert formal mathematical statements into their informal counterparts. Utilizing this newly constructed dataset, the authors further fine-tune various language models, achieving an acceptable output—with minimal adjustments needed—on 16-18% of statements when benchmarked against miniF2F and ProofNet.

**Strengths:**

Strengths

The Multilingual Mathematical Autoformalization (MMA) dataset was ingeniously synthesized using GPT-4, resulting in a substantial collection of 332,000 informal-formal pairings across multiple formal languages.
With its multilingual and multidomain composition, the dataset notably exceeds the size of the largest existing datasets in the field.
The research showcases improved outcomes when compared against established baselines.

**Weaknesses:**

Weaknesses

The integrity of the MMA dataset warrants further examination, as it was entirely auto-generated via GPT-4, potentially leading to mismatched cases. The paper would benefit from an expanded discussion on the discrepancies and the inclusion of findings from human evaluation.
The paper's innovative contribution seems incremental, largely relying on the automated capabilities of GPT-4 for dataset generation, which suggests a limited technical advancement.

**Questions:**

How do the authors check the quality of the dataset?

---

> ### Author Response · Authors · 2023-11-17
> **Official rebuttal**
>
> We thank the reviewer for their careful examination of our paper and insightful feedback. We are glad that the reviewer recognises the contribution of the dataset. Below we address the specific points raised by the reviewer.
>
> - Dataset quality
>     - We fully realise the importance of the dataset quality and hence conducted an in-depth examination of the dataset quality, posted in the overall comment: https://openreview.net/forum?id=QqdloE1QH2&noteId=Z8PeCPad8b
>     - To summarise the findings of the analysis:
>       - Around 75% of the examined 200 problems were informalised completely correctly by GPT-4.
>       - The failure cases are mostly due to ambiguity from the lack of appropriate context. And to ameliorate this issue is an exciting avenue for future research, possibly leveraging retrieval-augmented generation.
>
> - Contribution of the paper
>     - Our contributions are
>       1. Creating a large dataset
>       2. Training multiple language models on it with different settings
>       3. Evaluating these language models on held-out benchmarks manually to demonstrate that the dataset helps autoformalization
>       4. Conducting ablation experiments to show the multilinguality in formal languages benefits autoformalization.
>    - This will enable other researchers to conduct meaningful work on top of ours and advance science. As an investigative scientific work, we think that our contributions merit a broader audience.

---

> > ### Author Response · Authors · 2023-11-20
> >
> > Dear Reviewer d1G8,
> >
> > Did our rebuttal sufficiently address your concerns? Is there anything we can present that will convince you to increase your rating? We look forward to hearing from you.
> >
> > Many thanks,
> > Authors

---

### Author Response · Authors · 2023-11-17
**Overall rebuttal comment regarding MMA dataset quality**

- We want to thank all the reviewers for requesting a more detailed study regarding the MMA dataset quality. Since there is no automated metric that checks the correctness of the formal-informal alignment, we only presented 4 examples in the paper. However, we fully grasp the importance of this issue, and proceed to conduct a more detailed taxonomy of correct/incorrect MMA datapoints. Note that this is a preliminary taxonomy since we want to present an analysis during the discussion phase. We will refine it by inviting more formal mathematics experts to work on it in the final version.

- Before presenting the taxonomy, we would like to point out that the taxonomy is a direct analysis of the MMA dataset quality, while the improvement of the large language model on autoformalization is an indirect measure. With these two measures, we are convinced that the MMA dataset quality is high and can be of great use to autoformalization research.

- Taxonomy and analysis:
  - We randomly picked 100 statements from the Lean part of the MMA dataset and 100 statements from the Isabelle part of the MMA dataset. We then manually examine each formal-informal pairing carefully. We rate each problem on the following axes:

    - Correctness (whether the informalisation is completely correct): True or False
    - Judgment Confidence (the confidence of the assessment): 0 - 5
    - Hallucination (whether the informalisation includes content not intended in the formal statement): True or False
    - Misunderstanding concept (taking one concept in the formal statement for a different one): True or False
    - Incorrectly translating assumption: True or False
    - Incorrectly translating conclusion: True or False
    - Incorrectly translating type: True or False

  - We find that 67% of Lean statements are informalised correctly, and 81% of Isabelle statements are informalised correctly, with a total correctness rate of **74%**. Based on this statistic, we estimate the total correctness rate of the MMA dataset to be similar. We note that this is a relatively good correctness ratio in theorem proving: In [1], they found that even when only 25.3% of the autoformalization statements are completely correct, downstream applications were still able to benefit drastically from the parallel dataset.

  - Breakdown of failure reasons:
     We have a total of 52 statements that were informalised incorrectly. Here we show their failure reasons. Note that one incorrect statement can have multiple failure reasons.

    - Misunderstanding concept: 29
    - Hallucination: 8
    - Incorrectly translating assumption: 11
    - Incorrectly translating conclusion: 8
    - Incorrectly translating type: 12

  - We notice that the main reason for an informalisation mistake to happen is the ambiguity in the formal statement itself without proper context. This should be improved in the future work by incorporating relevant definitions and theorems.

   - Confidence
      - In rating these informalisations, we have an average confidence of 3.64 with a standard deviation of 1.19. Due to the confidence not being extremely high, we will open up the avenue for the community to contribute better annotations than ours.

  - We want to thank the reviewers again for raising the need for such an analysis as it drastically improves the quality of the paper and will incorporate this analysis in the camera-ready version, with the main conclusions in the datasets section and the detailed analysis in the appendix.

[1] Wu, Y., Jiang, A.Q., Li, W., Rabe, M., Staats, C., Jamnik, M. and Szegedy, C., 2022. Autoformalization with large language models. Advances in Neural Information Processing Systems, 35, pp.32353-32368.

---

### Meta-Review · Area_Chair_6tze · 2023-12-12

**Metareview:**

The paper presents an automatically generated Multilingual Mathematical Autoformalization dataset (=dataset of natural language and machine verifiable formalism pairs).

**Justification For Why Not Higher Score:**

Since the entire dataset is automatically generated via GPT-4, the paper introduces a technique rather than a dataset that is implicitly valuable.

**Justification For Why Not Lower Score:**

n/a

---

### Decision · Program_Chairs · 2024-01-16

Reject